# Liver Microenvironment Response to Prostate Cancer Metastasis and Hormonal Therapy

**DOI:** 10.3390/cancers14246189

**Published:** 2022-12-15

**Authors:** Alison K. Buxton, Salma Abbasova, Charlotte L. Bevan, Damien A. Leach

**Affiliations:** Division of Cancer, Imperial Centre for Translational & Experimental Medicine, Imperial College London, Hammersmith Hospital Campus, London W12 0NN, UK

**Keywords:** prostate cancer, liver, metastasis, microenvironment, niche, ECM, hormones, androgen

## Abstract

**Simple Summary:**

Prostate cancer patients with disease that has invaded the liver have the worst outcomes. Within the liver, cancer cells are exposed to a unique microenvironment of liver specific cells and proteins. In general, interaction between the microenvironment and cancer is known to provide cues which alter cancer cell biology and behavior. This review aims to summarize current knowledge about the microenvironment of the liver, what predisposes prostate cancer to move to the liver, how the liver responds to prostate cancer being there, and how the liver responds to current treatment strategies. We aim to provide insight into this under-investigated area of prostate cancer research as if we can understand why liver metastasis is associated with such poor patient outcomes, we will be better placed to address this.

**Abstract:**

Prostate cancer-associated deaths arise from disease progression and metastasis. Metastasis to the liver is associated with the worst clinical outcomes for prostate cancer patients, and these metastatic tumors can be particularly resistant to the currently widely used chemotherapy and hormonal therapies, such as anti-androgens which block androgen synthesis or directly target the androgen receptor. The incidence of liver metastases is reportedly increasing, with a potential correlation with use of anti-androgen therapies. A key player in prostate cancer progression and therapeutic response is the microenvironment of the tumor(s). This is a dynamic and adaptive collection of cells and proteins, which impart signals and stimuli that can alter biological processes within prostate cancer cells. Investigation in the prostate primary site has demonstrated that cells of the microenvironment are also responsive to hormones and hormonal therapies. In this review, we collate information about what happens when cancer moves to the liver: the types of prostate cancer cells that metastasize there, the response of resident mesenchymal cells of the liver, and how the interactions between the cancer cells and the microenvironment may be altered by hormonal therapy.

## 1. Introduction

Prostate cancer (PCa) is the fourth most commonly diagnosed cancer globally, with particularly high prevalence in western countries [1]. When confined to the prostate the disease is frequently relatively indolent; problems arise when the disease progresses and moves to other organs, with most PCa-related deaths being due to disease metastasis [2,3]. Developing from the luminal epithelial cells lining the glandular tubules of the prostate, PCa grows into the surrounding stroma, then can invade local surrounding organs (e.g., bladder, seminal vesicles) and lymph nodes, and in advanced cases, metastasizes to distal sites such as bone and/or visceral organs [4,5]. The liver is considered one of the most common sites for metastasis among solid tumors, with liver metastasis occurring in up to 50% of metastatic gastric/colon cancers, between 10–20% of metastatic melanoma cancers, 4–17% of metastatic lung cancer, 30–40% of metastatic pancreatic and 6–38% of metastatic breast cancers [6,7]. Historically, PCa-associated liver metastases are relatively uncommon—but when they do occur are associated with shorter overall survival times and are less responsive to treatments [8,9].

Since PCa cells proliferate in response to androgens, anti-androgen therapies are commonly the drug of choice for inoperable PCa treatment. These act at the level of the androgen receptor (AR), binding to it and preventing its activation, hence inhibiting androgen-responsive gene transcription [10]. Whilst this approach works well initially in the majority of cases, cancers often become resistant to such hormonal/anti-androgen therapy [11]. This is termed castration-resistant PCa (CRPC), and is associated with elevated levels of serum PSA, confirming progression is still driven by AR signaling as PSA is an AR target gene [3]. Second generation anti-androgens, such as bicalutamide and enzalutamide, are widely employed to more effectively target the AR signaling pathway and manage metastatic cases, but the issue of resistance still remains [12].

Anti-androgen therapies, which are given systemically, have also been associated with other effects in the liver [13], although few papers address this impact. One of the most common adverse effects seen with anti-androgens in the liver is liver injury onset [14]. In response to injury, the liver initiates both inflammation and fibrosis. Damage to the parenchyma causes release of paracrine factors, which recruit immune cells and activate stellate cells, which in turn release a plethora of factors causing recruitment of more immune cells and activated stellate cells, myofibroblasts, neo-angiogenesis, and deposition of a dense extracellular matrix (ECM) [15]. In time this process resolves and liver regeneration occurs, but prolonged damage responses can be detrimental to liver function, and the prolonged fibrotic response can also create a microenvironment or pre-metastatic niche which is favorable/supportive for metastatic seeding and growth [16,17,18,19,20]. Furthermore, in those colorectal and breast cancers that metastasize to the liver, there are several changes associated with ECM interactions that are not seen in tumors that metastasize to different organs [21], indicating that the liver ECM environment may be selective for certain types of metastatic cells.

The vital role of ECM in disease progression is also seen in the primary site [22,23] where in PCa the ECM is known to regulated by AR [24]. AR signaling in the stroma of the microenvironment has been shown to affect cancer cells [25]. Previously, stromal AR was shown to be vital for prostate growth and development, whilst epithelial AR was known to be responsible for the androgen-dependent synthesis of epithelial secretory proteins [26,27]. More recently, AR signaling in the cancer stroma was shown to impact patient outcomes by inhibiting cancer cell invasion, however, the mechanisms behind the roles stromal AR plays in disease progression are not fully understood [24].

The changes to the tumor microenvironment and cells therein highlight its importance in cell invasion and metastasis. The aim of this review is to collate the current knowledge of what types of PCa metastasize there, what happens to the liver when PCa invades, and what associations exist with current therapeutic options.

## 2. Metastasis to the Liver

The liver is a large organ which primarily filters blood coming from the digestive tract, but also has critical roles in metabolism and secretion. At a microscopic level, the liver is composed of liver lobules, which are roughly hexagonal in shape (Figure 1A). At the outside of each segment of each lobule are hepatic triads comprised of an artery, a vein, and a bile duct. These flow toward the center of the lobule, where there is a central vein. Between the triad and central veins are sinusoids through which the mixed veinous and arterial blood flows, each sinusoid is composed of specialized parenchymal cells called hepatocytes and lined with endothelial cells. Between the parenchymal hepatocyte cells and the endothelial cells is the space of Disse, in which mesenchymal and inflammatory cells reside.

In the primary site, cancer cells invade through the stroma and escape the primary site via intravasation into the vasculature, where they circulate until they leave the vasculature (extravasation) and colonize the liver. In the liver, cancer cells either undergo cell death, or they can remain in a dormant state, or they proliferate and form micro-metastases until growing enough to become a macroscopic metastatic lesion (Figure 1B).

As previously summarized, many solid tumors metastasize to the liver [6]. In a study of 74,826 PCa patients with metastatic disease, 84% of metastases were found in the bone, whilst liver metastasis accounted for 10.2% [28]. In a retrospective analysis of the SEER dataset, liver metastasis occurred in around 3% of metastatic PCa (20,034 men, [29]), whilst in clinical trials for advanced disease, liver metastases account for between 12–30% [30,31].

## 3. Prostate Metastasis to the Liver

Liver metastasis is associated with some of the worst clinical outcomes compared with other sites [32,33]. Out of all the patient with liver metastasis, those which originated from the prostate have the second worse survival (behind testicular cancer) [34]. Kelly et al. (2012) report that patients with liver metastases had a median survival of 14.4 months compared with 22.2 months in patients with non-liver metastases [35]. A meta-analysis of PCa samples in the SEER dataset (n = 10,777) confirmed that the presence of liver metastasis in patients associated with the worst cancer specific and overall survival [36]. A full summary of the effects of visceral metastases on patient outcomes in different patient cohorts is shown in Table 1.

In general, AR status of PCa cells has an inverse correlation with neuroendocrine features (Figure 2B). Publicly available data suggest that AR-negative and neuroendocrine disease are most often found in liver metastases (Figure 2A), indeed anecdotally, liver metastases are usually presumed to all be AR-negative and neuroendocrine. However, a mixture of AR-positive or -negative neuroendocrine phenotypes, or indeed AR-negative non-neuroendocrine PCa, have been seen, with varying degrees of heterogeneity within each tumor [37]. The data also suggest that in PCa, liver metastases can be either composed of AR-negative/neuroendocrine disease or AR-positive/adenocarcinoma disease, and in some cases potentially a mixture of the two types (Figure 2C). Patients with liver metastases have can have high circulating PSA levels, indeed in meta-analysis of multiple studies, relating to 8820 men in total with mCRPC, patients with liver metastasis (n = 752 men) had the highest median levels of serum PSA of all the metastatic sites tested [8].

Compared to other metastatic sites for PCa, liver metastases reportedly have the highest fraction of genomic alterations and there are also potential liver specific alterations such as MYC amplification, PTEN deletion or PIK3CB amplification [33]. PCa may undergo site specific transcriptional changes, whereby the tumors adapt to survive and thrive in that specific micro-environment, and it may be that the changes in metabolism pathways are especially important to PCa that have located to the liver [38,39]. All three of these proteins have been linked with metabolism: PTEN with glycolysis and mitochondrial activity [40], MYC regulates genes involved in biogenesis of ribosomes and mitochondria, and glucose/glutamine metabolism [41], while PIK3CB is associated with metabolism of cholesterol, triglycerides, and sugars [42,43].

In other cancers that invade the liver, measurable changes have been noted in liver function serum markers alanine transaminase (ALT), aspartate transaminase (AST), total bilirubin (TBIL), gamma glutamyltransferase (GGT), alkaline phosphatase (ALP), albumin (ALB), and carcinoembryonic antigen (CEA) [44,45,46]. Serum markers of liver function are also dysregulated when PCa invades the liver [47]. In an analysis of 1281 men with PCa, increased serum levels of AST and/or LDH, or a decreasing serum hemoglobin, were associated with an increased probability of PCa metastasis to the liver [47].

Liver metastases express the epithelial-characteristic adhesion protein E-cadherin at a level that is equivalent to or higher than matched primary tumors [48,49,50]. This is different to what is observed at other metastatic sites, where E-cadherin is more often expressed at low levels, and may suggest that tumors that metastasize to the liver have a different relationship to other cell types and the local microenvironment in terms of cell–cell and cell-ECM interaction. The liver microenvironment is considered vastly different to other metastatic sites, such as the bone [51] and it has been postulated that the microenvironment of the liver plays a role in determining the “types” of PCa that will successfully colonize the liver and how lethal they will be [51,52]. In other solid tumors that metastasize to the liver, proliferating cancer cells in the liver form subclinical micro-metastases in the periportal regions of the liver lobule, and their growth and survival is supported by the host parenchymal and stromal cells [53]. In liver spread of colon, pancreatic, and breast cancers, the presence and activation of resident stellate cells is important in preparing a microenvironment that propagates metastatic seeding [54] and growth [55,56,57,58]. The ability of stellate cell activation to influence metastatic growth is reported to involve their ability to influence fibrosis and ECM deposition/stiffness [58,59,60]. It will be interesting to confirm how the liver microenvironment actually responds to PCa.

## 4. Anti-Androgens and Patients with Liver Metastases

As stated earlier, PCa growth is dependent on androgens, which are male sex hormones that work through the androgen receptor (AR) [62]. The AR is a nuclear transcription factor activated by testosterone or dihydrotestosterone (DHT) and is a key driver in PCa development and progression [63]. Androgen deprivation therapy (ADT) is common treatment for recurrent or metastatic PCa [64]. These therapies work by either reducing circulating testosterone levels or inhibiting androgen binding to AR; both approaches reduce AR activity [65]. The latest generation of anti-androgens includes enzalutamide, and more recently darolutamide and apalutamide [10,66], and these are generally effective in lowering PSA levels, reducing tumor burden, and increasing patient survival [11,67]. These have varying success rates, dependent on when in the disease progression they are administered and the status, e.g., whether the primary site is intact [10,68].

One of the principal factors that determine the responsiveness and overall success of anti-androgen therapies is the site of metastasis. The poor clinical outcomes associated with visceral metastases, in particular liver metastases [30] (Table 1), could be in part due to the reported unresponsiveness of liver metastases to multiple types of anti-androgen therapy [8,9,69,70,71]. There have been studies reporting success of anti-androgen therapies in liver metastasis cases [72], however, many suggest this effect is short-lived [9]. A possible explanation for this may be that liver metastasis is considered to be a late event in disease progression, linked to neuroendocrine tumor characteristics including lack of AR expression (see above), and therefore represents an aggressive subset which does not respond well to therapy [69]. However, as discussed, there are cases of AR-expressing adenocarcinoma metastasizing to the liver, which could be responsive. Furthermore, liver metastases can occur in patients who do not exhibit bone metastases, with limited differences in survival between patients with liver only metastasis compared to patients with liver plus other metastatic sites [8,28].

Reports also suggest that cases of liver metastases are increasing and this could possibly be linked to the use of anti-androgen therapies [5,31,73]. This could be due to anti-androgens exerting selecting pressure that favors cancer cells that home to the liver, or selecting for neuro-endocrine type tumors [74]. This may also suggest anti-androgen therapies are having a more significant impact on the liver itself than previously thought—as discussed below, there is evidence for links with fibrosis.

**Table 1 cancers-14-06189-t001:** Summary of clinical trials to assess efficacy of anti-androgen therapies and response of patients with visceral metastasis.

Paper	# of Patients	Tumor Type	Treatment	Outcome
Conteduca et al., 2015 [75]	265	CRPC	Abiraterone	VM linked to reduced OS
Goodman et al., 2014 [76]	1195	CRPC	Abiraterone acetate or placebo	VM associated with reduced PFS and OS in both groups
Poon et al., 2016 [77]	110	mCRPC	Abiraterone acetate	Chemotherapy naïve-VM reduced OS and PFS Chemotherapy received-VM no sig. dif.
Moschini et al., 2016 [78]	1011	LN + ve PCa		VM had poor OS VM had greater HR
Gandaglia et al., 2015 [79]	3857	mPCa		VM alone or with BM had worse OS and PFS than BM or LN
Conteduca et al., 2016 [80]	193	mCRPC	Enzalutamide	VM increased HR but not significant
Armstrong et al., 2007 [81]	1006	mCRPC	Docetaxel, mitoxantrone, prednisone	VM and multiple sites had higher HR
Pond et al., 2014 [32]	1006	mCRPC	Docetaxel, mitoxantrone, prednisone	Liver or lung had worse OS than BM
Terada et al., 2016 [82]	329	mPCa	Enzalutamide	VM increased CRPC development more than BM, lower PFS than BM or LN
Shiota et al., 2014 [83]	97	CRPC	Docetaxel and Prednisone	VM has worse OS and PFS
Loriot et al., 2013 [84]	307	mCRPC	Enzalutamide (previous docetaxel)	OS increased in lung and liver, bigger effect in lung
Loriot et al., 2017 [85]	1199	CRPC	Enzalutamide	OS increased in lung and liver, bigger effect in lung
Penson et al., 2016 [86]	396	CRPC	Enzalutamide vs. Bicalutamide	Enzalutamide had better PFS
Davies et al., 2019 [87]	1125	mPCa	Testosterone and Enzalutamide	Better OS and PFS with testosterone/enza vs. control
Eisenberger et al., 1998 [88]	1378	mPCa	Flutamide	No sig. dif. in OS

CRPC = castration resistant prostate cancer, mCRPC = metastatic castrate resistant prostate cancer, LN + PCa = lymph node positive prostate cancer, mPCa = metastatic prostate cancer, LHRH = luteinizing hormone-releasing hormone, VM = visceral metastasis, PFS = progression free survival, OS = overall survival, BM = bone metastasis, HR = 5-year hazard ratio, Enza = Enzalutamide.

## 5. The Influence of Microenvironment on PCa Progression and Its Relationship with AR

The influence of the microenvironment is seen throughout all stages of cancer development and progression and therefore, therapies to target cancers must consider the effect of (and on) the tumor microenvironment.

Prostate development is reliant on the interactions between epithelial cells and the surrounding stroma in the microenvironment. The prostatic stroma is composed of several non-malignant cell types such as fibroblasts, smooth muscle cells, endothelial cells and immune cells [89]. In cancer, the components of the stroma also undergo a type of transformational change and the signaling between cancer cells and the microenvironment is altered [90]. In the primary site, the prostate microenvironment changes from the benign context of smooth muscle cells and fibroblasts, to the cancer-associated microenvironment largely composed of fibroblasts with an activated phenotype, termed cancer associated fibroblasts (CAFs) [91]. CAFs are spindle-shaped cells that are derived from the fibroblasts present in the normal microenvironment [23,25,92], but differ from them by producing elevated levels of collagen and ECM proteins and upregulating the secretion of pro-tumorigenic factors [93] to facilitate tumor growth, invasion and metastasis [94]. In addition, this CAF microenvironment secretes growth factors which promote angiogenesis, alter ECM architecture, and accelerates fibroblast proliferation [95]. Without these changes to the microenvironment, cancer cells cannot invade and metastasize.

Androgen receptors are not just expressed in cancerous epithelial cells, but in multiple cell types of the prostate stroma [89,96]. Stromal AR is well documented to be essential for normal prostate growth and development [97]. In PCa, reduced AR levels in the stroma is frequently associated with poor clinical outcomes [24,98,99,100,101,102,103]. Moreover, low stromal AR expression is linked to tumor resistance to ADT [103] and relapse in PCa patients [100]. This is the opposite of AR’s effect in epithelial cells, where high levels of AR were found to be associated with a more aggressive disease phenotype [24]. A meta-analysis of protein markers in PCa confirmed that low or no AR expression in the PCa stroma associated with worse patient outcomes, and stromal AR is in fact one of the only markers that consistently associates with progression [104,105]. Niu et al. (2008) also suggest stromal AR is essential for cancer initiation and growth in mouse xenograft models using WPMY cells, with reduced AR in the stroma found to be effective at reducing tumor growth initially; they then also suggest low stromal AR suppressed metastasis [106], although these cell line data are not supported with clinical findings. This may suggest the effect of AR expression is dependent on the stage and progression of the disease, and AR may play a different role as the disease changes. Given the significance of AR in stroma of the primary site, the effect of anti-androgens on the metastatic microenvironment warrants further consideration.

The role of AR in the metastatic microenvironments is generally under-investigated, this is particularly true for the liver. A number of articles have suggested that the liver is responsive to androgens and expresses AR both in the nucleus and cytoplasm (only determined in whole tissue extracts and hepatocytes) [107,108,109,110,111]. Single cell RNA-sequencing data sets show detectable AR RNA heterogeneously expressed throughout the liver, but mainly in hepatocytes, vasculature and mesenchymal cells (fibroblasts/stellate cells) although there are also a few immune cell types which express AR (Figure 3).

There is some data suggesting an association between hepatic AR and reduced immune infiltration [112] and enabling glycolysis and metabolism [113]. Additionally, androgens may have a role in regulating the secretion of cytokines and growth factors, such as TGF-ß and VEGF, by hepatocytes [113,114]. Importantly, TGF-ß is integral to fibrotic responses, and is also known to be increased in PCa liver metastases [51,115], and to promote invasion and metastasis, all potential reasons why cancer cells metastasize to and grow in the liver [116]. CAFs, detectable in prostate liver metastases [117], potentially work in combination with host mesenchymal cells (stellate cells), which are recruited intra-metastatically and incorporated to form a stroma which releases growth factors and ECM proteins to support cancer cell growth [53]. It is not yet known whether AR signaling in these two cell types will resemble what we have previously reported for AR in prostatic fibroblasts [24,25]. Despite this emerging research, the role of androgen signaling in the liver metastatic microenvironment and its involvement in influencing local cancer cell biology is not well defined.

## 6. How the Liver Responds to Anti-Androgen Therapies

It is underappreciated that all cell types throughout the body express some form of nuclear/sex hormone receptor [120]. Cell comprising the liver express all steroid receptors, including AR, and the liver appears to be an androgen responsive organ [121,122], which may be further suggested by the gender imbalance in all types of liver disease and fibrotic states [108,121,123,124,125]. In a study of 117 men, higher levels of testosterone were associated with lower serum ALT and AST [126]. There have been suggestions that androgen metabolism and liver cirrhosis are linked [127]. Indeed androgen (and estrogen) levels have been associated with development of liver disease [128]. Furthermore, prohibitin (PHB), an AR co-repressor protein [129,130,131], is associated with liver injury and cancer [132], and importantly in a liver specific PHB knock out mouse model, there was upregulation of genes involved in fibrosis [133]. Sex also appears to influence the incidence of liver metastasis, with men twice as likely to present with liver metastases than women [134]. Interestingly in a mouse model of diethylnitrosamine (a hepatotoxicant and hepatocarcinogen)-induced formation of hepatocellular carcinoma (HCC), this treatment was also associated with fibrosis and accumulation of AR positive mesenchymal cells and immune cells [112]. This may support the previously mentioned hypothesized link between AR and fibrotic responses in the liver.

Liver injury associated with anti-androgen therapies has been reported in the literature since as early as the 1940s [135]. Analysis of the SEER database indicated that in men receiving some form of ADT, there was a significant association with subsequent diagnosis of liver disease, including non-alcoholic fatty liver disease, cirrhosis and necrosis [136]. Liver injury was associated with anti-androgen therapies involving first-generation antiandrogens such as Flutamide, albeit rarely [14]. The same report also identified a case of liver toxicity associated with Nilutamide administration. Yun et al. (2016) also reported liver injury associated with (second generation antiandrogen) bicalutamide treatment [137], with histology demonstrating acute intrahepatic cholestasis suggestive that the injury was caused by anti-androgen administration. In large trials assessing the use of abiraterone, which is a steroid synthesis inhibitor that has also been shown to have antiandrogen effects [138], there have been trends of liver enzyme increases indicative of damage/reduced function. In a PCa cohort of 1917 patients, the administration of abiraterone saw the percentage of patients with increased ALT and AST levels rise from below 1% to 6% (53 patients) [139]. In the COU-AA trial of prednisone alone or in combination with abiraterone, there was a small increase in abnormal liver function tests, with the number of patients with increased AST and ALT doubling from, respectively, 5% and 4.8% of patients with prednisone alone to 13.3% and 12% in patients with prednisone and abiraterone, albeit this involved a small number of patients and the effect was not quite significant [140,141]. This trend was observed in other trials involving abiraterone combinations, where small increases in the number of patients with high levels of LDH, AST and ALT were observed [12,142]. The increase in liver enzymes can resolve with time [143], but the effect of this increase and potential damage to the liver is unknown. So, whilst not all patients will have liver damage events in response to anti-androgens, it may be worthwhile monitoring those patients that do for subsequent metastases.

## 7. Potential for the Formation of a Pre-Metastatic Niche

A major concern of tissue response to injury is the possible formation of a pre-metastatic niche. A pre-metastatic niche is a microenvironment which is suitable/optimal for colonization by circulating tumor cells [144]—the “soil” in the ‘seed and soil’ hypothesis [145,146]. In the liver, changes to the stromal cells and ECM in response to liver injury produce an environment conducive for cancer cell seeding [146] and patients with liver fibrosis have an increased risk of developing liver cancer [147]. Liver injury and fibrosis have also been associated with subsequent metastases, with the fibrotic environment providing a niche for cancer growth [18,19,148,149]. Liver injury creating a pre-metastatic niche has been reported in pancreatic and colorectal cancer [18,19,20,150,151].

The fibrosis/wound healing activated in response to injury in the liver is similar to the changes observed in the microenvironment of cancer metastases discussed earlier (Figure 1). Hepatic stellate cells are essential for the liver’s response to injury, becoming activated and transforming into myofibroblast-like cells which produce ECM, culminating in wound healing responses to protect the liver, but simultaneously reducing tissue structure and function [152]. There is also an accrual of fibroblasts, which also aid ECM production and accumulation [153]. The TGF-ß and ECM produced by activated stellate cells has been reported to increase cancer invasion and proliferation in pancreatic and colon models, while activation of stellate cells is reported to activate growth of dormant breast cancer micro-metastases [154]. In colorectal cancer, patients with a fibrotic liver had a four-fold increased risk of liver metastasis compared to those with a normal liver [19]. In terms of mechanisms involved, we may gain insight from other metastatic sites. For example, in lung metastasis, cases have been shown to be more severe where increased fibronectin and stiffening of the ECM are present, which can nurture growth by overriding tumor suppressor activity [155]. Fibronectin produced by activated stellate cells has been reported to cause recruitment of bone marrow-derived immune cells, creating an environment that allowed metastatic growth of pancreatic cancer cells in animal models [156]. Activated stellate cells have also been reported to secrete CCL20 and increase fibronectin deposition, promoting colorectal cancer cell line metastasis to the liver in mouse models [54]. Colorectal metastasis has also been reported to be enhanced by tenascin C produced by activated stellate cells [157]. Activation of stellate cells has been reported to alter the ECM they produce, increasing collagen and periostin, to promote metastatic growth of pancreatic cells [158]. Alterations in ECM components activate intracellular signaling pathways, such as the Akt pathway, in cancer cells to promote invasion, seeding, growth, and survival [159]. Importantly, it should be noted that the ECM is able to affect the sensitivity/responsiveness of cancer cells to therapy [160]. In a meta-analysis of five PCa studies (434 samples), genes involved in focal adhesion signaling were significantly altered in liver metastases [38], indicating that cell-ECM adhesion/interactions have an important role in cancer metastases. Additionally, the activation of stellate cells also causes the secretion of a milieu of growth factors and cytokines (PDGFs, HGF, TGF-β, SDF-1, VEGF, PGF, FGFs, CXCLs) which can also promote cancer proliferation and development of micro-metastases [161].

Given the responsiveness of the liver to androgens, there is potential that anti-androgen therapies may contribute to reported liver injury associated with administration. A worrying potential result of this side-effect may be the creation of a microenvironment more hospitable to metastatic seeding and growth in certain patients.

## 8. Conclusions

Metastasis to the liver has one of the worst cancer outcomes yet there is relatively little known about liver metastases, the role of the liver microenvironment, and why prognosis is so poor. It is important to investigate these aspects of the disease given the reported increase in cases. Current treatment options, including anti-androgen therapies, have limited success rates in liver metastasis cases. When comparing the effectiveness of anti-androgen therapies, patient with liver metastases were significantly less responsive than patients with only bone metastases [8,31]. In some patients, anti-androgen therapy (perhaps particularly flutamide and bicalutamide) can induce liver injury [14,137]. In response, the liver recruits cells to form scar tissue—a similar response to what is observed in the formation of a pre-metastatic niche. A concern is this side effect may result in the area becoming more attractive to cancer cell invasion [146]. This may need to become a research priority to ensure treatment does not result in disease progression to a lethal stage. The changes seen to the liver microenvironment are poorly characterized; further research is required to determine why prostate cancer often spreads here and why such poor clinical outcomes are associated with liver metastasis.

## Figures and Tables

**Figure 1 cancers-14-06189-f001:**
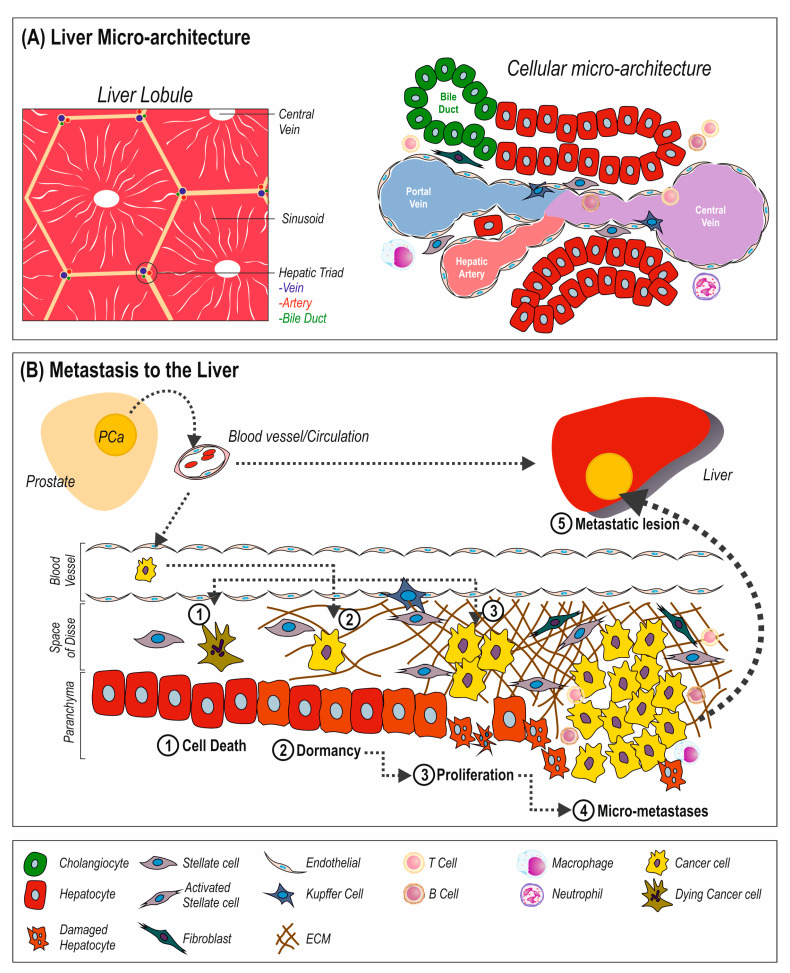
(**A**) Liver Lobule Cellular Microenvironment. The liver is comprised of hepatic lobules, which are the microscopic hexagonal subunits of the liver. These consist of a central collecting vein, hepatocyte lined sinusoids leading to hepatic triads, a collection of three ducts; hepatic artery, portal vein, and bile duct. Between the endothelial cells and either hepatocytes or cholangiocytes/bile ducts, is the space of Disse, a region where mesenchymal cells reside, such as stellate cells, dendritic cells/immune cells, and fibroblasts. (**B**) Metastatic colonization of the liver. Escaping from the primary site, cancer cells invade the circulatory system, through which they navigate to the liver. Once extravasated from hepatic blood vessels and in the liver, cancer cells can undergo cell death or they can remain dormant. From dormancy, cancer cell proliferation can be activated, allow for the formation of micro-metastases, which with continued proliferation become macroscopic metastatic lesions. There is an associated change in the resident cells, with damage to the hepatocytes, increase in ECM due to activation of stellate cells and recruitment of fibroblasts. There is also a potential influx of immune cells.

**Figure 2 cancers-14-06189-f002:**
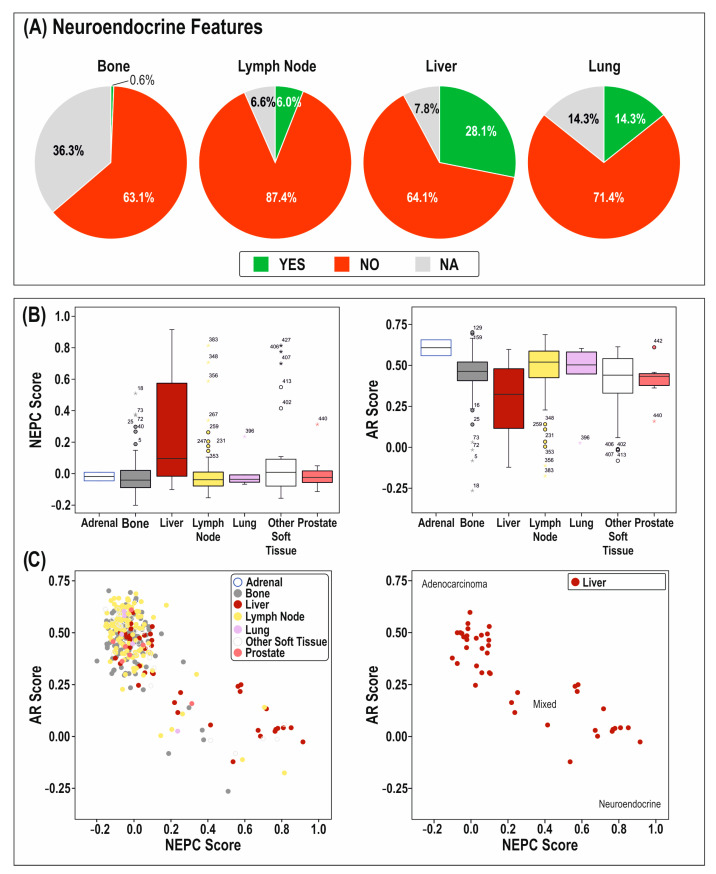
Characteristics of prostate cancer cells in the liver. A summary of data from Stand Up to Cancer (SU2C) database showing the cellular characteristics of prostate cancer cells that have metastasized to the bone (n = 160), lymph node (n = 167), liver (n = 64), and lung (n = 7). (**A**) The per-centage of metastatic tumors with pathologist defined neuroendocrine features. Green = metastases with neuroendocrine features, red = no neuroendocrine features, and grey = no characterization available. (**B**) Analysis of (i) neuroendocrine prostate cancer (NEPC) gene scores and (ii) androgen receptor (AR) activity gene scores in different metastatic sites. (**C**) Comparison of NEPC (horizontal) and AR (vertical) gene scores in (i) different metastatic sites, and (ii) the liver alone. Analyzed using publicly available data [61].

**Figure 3 cancers-14-06189-f003:**
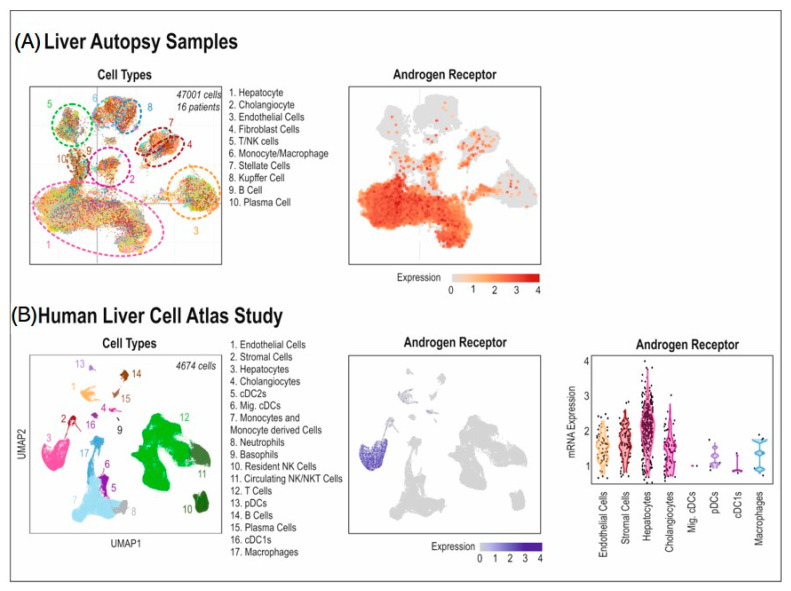
AR expression in the cells of the liver. Single cell RNA-seq analysis of human liver from two different studies. (**A**) Liver autopsy samples [118] of 47,001 cells in total from 16 patients. (i) UMAP graph of single cells collected from different patients’ livers. Each sample is colored by which patient they come from, and surrounded by a dotted line indicating which cell types compose each group. (ii) Analysis of AR expression in each single cell, with expression scaled from grey (no expression) to increasing shades of red (increasing levels of AR expression). (**B**) The Human Liver Cell Atlas Study (GSE192742 [119]), comprised of 4647 cells from human liver. (i) UMAP graph showing the cell-types present in this sample. (ii) Analysis of AR expression in each single cell, with expression scaled from grey (no expression) and deepening shades of blue (increasing levels of AR expression). (iii) Violin plots of AR expression in a sub-set of the cell-types detected in the study.

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
