# Peer review of "Liver Microenvironment Response to Prostate Cancer Metastasis and Hormonal Therapy"

_cancers, 2022, doi:10.3390/cancers14246189_

Round 1

Reviewer 1 Report

                Although the incidence of liver metastases in prostate cancer is increasing, there is a relative dearth of studies examining the biology of disease progression in the liver. In this review, Buxton et al. examine how the liver microenvironment contributes to disease progression and therapeutic response in metastatic prostate cancer.  They also discuss how the microenvironment may be altered by hormonal therapy.

                This is a well-written, fairly-balanced review that summarizes the current literature. The authors present several relatively under studied and potentially controversial topics. Highlights include the role of AR in the metastatic microenvironment. The possibility that the incidence of liver metastases could be due to the use of anti-androgens has clinical significance.  The hypothesis that anti-androgens may result in liver injury and thus provide a more favorable environment for liver metastases deserves further study.

The figures are excellent and contribute to reader understanding.  The Reference list is exhaustive and well-balanced.

                There are only a few minor suggestions. The Simple Summary should be rewritten to avoid word repetition. The authors also might want to consider citing specific key findings in the Abstract.

                Lines 16 and following.  A brief review (1 o 2 sentences) of the mechanism of action of antiandrogens vs agents such as abiraterone would be useful here, although this is discussed further in the article.

                Are the differences in the box plot scores in Figure 2B and C statistically significant?

Author Response

We thank the reviewer for their positive feedback. We have included the suggestions of reworking the simple summary to make it less repetitive. We have also added an additional line to the abstract to broadly introduce the actions of anti-androgens. 

Reviewer 2 Report

This is a nicely written and well-documented review article on a timely topic: liver metastasis from prostate cancer. The authors present an overview of the field. They discuss genetic and phenotypic aspects of prostate cancer metastasis and the role of microenvironmental liver-specific factors and provide helpful insights for future lines of investigation in this emerging area of research.  

Author Response

We thank the reviewer for their resposne.

Reviewer 3 Report

The article by Buxton et al. submitted to Cancers is a review focused on liver metastasis in prostate cancer, with a particular interest in the microenvironment response.  The review describes multiple aspects of liver metastasis, including an overview of liver architecture and pathology before diving into understanding how stromal cells of the liver are (or may be) involved in the metastatic process.  An intriguing section follows, where they nicely summarize recent studies that suggest approved therapies such as androgen ablation can profoundly affect the liver, and they speculate how this response would impact disease progression.  The authors have also included their own analyses of publicly available databases to characterize AR status and neuroendocrine features in the liver. 

The liver is a lesser-known organ site for prostate metastasis.  As such, this review helps draw attention to an understudied area, and although many facets of liver metastasis are not understood, the authors have assembled a nice collection of studies specifically related to prostate-liver metastasis where possible, and they draw on published literature from other cancers as needed in an appropriate manner.  Highlighted below are some comments and suggestions that would strengthen the review prior to final acceptance.

Comments and Suggestions: 

·       Typo line 63: “stellates” likely should be stellate

·       Figure 1 Legend:

o   Consider mentioning/defining “Space of Disse” in the text legend for Figure 1.  It appears in the text and is indicated on the figure but including in the legend would allow for easier reference.

o   Comma is not needed after vessels in the sentence “Once extravasated from hepatic blood vessels,

o   Consider removing the word “itself” from the sentence “Once extravasated from hepatic blood vessels and in the liver itself, cancer cells can undergo…” for conciseness

o   Consider adding comma after “dormancy” in sentence: “From dormancy, cancer cell proliferation can be activated…”

·       Section 3 “Prostate metastasis to the liver”, starting at line 107:  the authors should consider adding median survival of patients with metastases in other organs, such as bone, if the data are available.  The authors provide a nice comparison of occurrence of metastases in other organs in the prior section, and including similar comparisons here would add continuity.

·       Typo line 114: “wort” should likely be “worst”

·       Line 117: Here neuroendocrine disease is mentioned as being inversely correlated with AR in prostate cancer and that it is often detected in liver metastases. References for this information should be added.

·       Line 129:  As presently written, it is unclear whether the genomic alterations that cancer cells that colonize the liver referred to before the semicolon are different from MYC amplification, PTEN deletion, and PIK3CB amplification described afterward.  If so, the reference is missing, and some examples should be given.  Overall, this sentence could perhaps be rewritten more clearly.

·       Line 172:  Consider rephrasing to read as “The latest generation of anti-androgen includes enzalutamide, and more recently darolutamide and apalutamide [10,55], and these are generally effective in lowering PSA levels, reducing tumor burden, and increasing patient survival [11,56].”

·       Line 183:  The authors make a point that liver metastasis is considered a late event in disease progression, which raised a few questions that they may wish to discuss.  For example, do liver metastases newly arise in patients with existing metastatic disease in other organs, or do some patients present at clinic solely with liver metastases?   

·       Line 187: removing “theoretically” would strengthen this statement

·       Line 212:  typo “secrets” should likely be “secretes”

·       Line 235:  Here it is not clear which cells in the liver are expressing AR in the nucleus and cytoplasm.  This needs some clarification.

·       Line 241:  Associations between hepatic AR and immune infiltration and control of metabolism are mentioned, but it is not mentioned whether these are positive or negative.  A few details to clarify should be added here.

·       Line 255:  The authors state here that all cell types throughout the body express some form of nuclear/sex hormone receptor.  A reference supporting this should be added.

·       Line 256:  Consider beginning this sentence as follows: “Cells comprising the liver express all steroid receptors…”

Author Response

We would like to thank the reviewer for the positive review and have attempted to alter the submission based on their suggestions. We apologise for the various typos, and these have been corrected.

  • In regards to figure 1, we have added clarification of the space of Disse to the text, and adjusted the other text within the legend as suggested.
  • In regards to the text in section 3, we reworded to highlight that the comparisons was between patients with liver metastases vs patients with non-liver metastases.
  • We adjusted line 129 to make it clearer about the relevant genetic alterations.
  • We reworded line 172 as suggested.
  • We added clarification about the expression of AR in the liver from line 235, with added references.
  • We added text to inform about the trends between hepatic AR and metabolism and immune infiltrate.
  • We added references supportin the sentence about systemic expression of nuclear/sex hormone receptors